# Understanding Alternatives to Tobacco Production in Kenya: A Qualitative Analysis at the Sub-National Level

**DOI:** 10.3390/ijerph17062033

**Published:** 2020-03-19

**Authors:** Madelyn Clark, Peter Magati, Jeffrey Drope, Ronald Labonte, Raphael Lencucha

**Affiliations:** 1Faculty of Medicine, School of Physical and Occupational Therapy, McGill University, 3630 Promenade Sir William Osler, Montreal, QC H3G 1Y5, Canada; madelyn.clark@mail.mcgill.ca; 2School of Economics, University of Nairobi, Nairobi 30197, Kenya; ptrmagati@gmail.com; 3Economic and Health Policy Research, American Cancer Society, Atlanta, GA 30303, USA; jeffrey.drope@cancer.org; 4School of Epidemiology and Public Health, University of Ottawa, Ottawa, ON K1N 6N5, Canada; rlabonte@uottawa.ca

**Keywords:** tobacco control, public policy, agriculture, tobacco farming

## Abstract

Tobacco is a key cash crop for many farmers in Kenya, although there is a variety of challenges associated with tobacco production. This study seeks to understand alternatives to tobacco production from the perspective of government officials, extension officers, and farmers at the sub-national level (Migori, Busia, and Meru) in Kenya. The study analyzes data from qualitative key-informant interviews with government officials and extension officers (*n* = 9) and focus group discussions (FGDs) with farmers (*n* = 5). Data were coded according to pre-identified categories derived from the research aim, namely, opportunities and challenges of tobacco farming and alternative crops, as well findings that illustrate the policy environment that shapes the agricultural context in these regions. We highlight important factors associated with the production of non-tobacco agricultural commodities, including the factors that shape the ability of these non-agricultural commodities to serve as viable alternatives to tobacco. The results highlight the effect that several factors, including access to capital, markets, and governmental assistance, have on farmer decisions. The results additionally display the structured policy approaches that are being promoted in governmental offices towards agricultural production, as well as the institutional shortcomings that inhibit their implementation at the sub-national level.

## 1. Introduction

Small scale export-driven agricultural production in Kenya, as in many other African countries, is both a driver for and a challenge to economic development. Agriculture is the mainstay of Kenya’s economy, accounting for at least 30% of the Gross Domestic Product (GDP). Smallholder farming accounts for roughly 75% of agricultural production [1], and continues to face several persisting challenges ranging from the year-to-year unpredictability of weather (which is growing worse with climate change in some regions) and markets, to the need for and often absence of infrastructure and other supports to small-scale growers to produce and bring to market quality crops [2]. Kenya remains a major agricultural producer and exporter for commodities such as coffee, tea, Asian vegetables, and tobacco, the latter of which is the focus of this article. Government data indicate that tobacco growing contributes 0.03% of Kenya’s GDP while the industry as a whole accounts for about 7% of the GDP. At least 55,000 farmers grow tobacco on 13,500 hectares of land mainly in the Western and South-Eastern parts of the county [3,4]. Although overall contribution to the Kenyan economy is relatively small, tobacco is an important economic activity in the regions where it is farmed. 

Although agricultural policy is established at the national level, the landscape of agricultural governance in Kenya has experienced important changes with the adoption of the new constitution in 2010. Sub-national governments are now charged with farm-level services and supports, and management of major agricultural production including dairy, sugar, and cotton, among others [5]. Some major cash crops such as coffee remain under the mandate of national agencies, given the extensive production across counties [5]. Despite these shifts, tobacco remains an unscheduled crop in Kenya, meaning that the government does not monitor or support tobacco production. Issues pertaining to alternatives to tobacco production, however, are situated within a wider market and governance context, and it is this context of financial, technical and market supports that farmers make decisions about whether to grow tobacco or to pursue alternative livelihoods [6]. 

The structural arrangements in Kenya supporting tobacco production have shifted over the decades, most markedly in the 1980s when British American Tobacco (BAT) implemented a contract farming regime. This regime enlisted smallholder farmers in contractual arrangements with the company to structure the supply of tobacco leaf, while providing access to inputs, loans and markets for the farmers. At the same time BAT rolled out a promotions campaign in Kenya leading to the doubling of tobacco consumption within the country [7]. This period also saw a sharp increase in production from 209 tons of tobacco leaf in 1975 to 4034 tons in 1982 [7]. Tobacco production continued to rise until peak production in 2005 (25000 tons) and has fluctuated since then with total production of 9711 tons in 2018, the last year with reported data [8]. Prices of tobacco leaf and farmers’ income have also fluctuated over the same period.

Despite fluctuations in production levels, prices, and farmer household income, an oft-touted narrative about the economic benefits of tobacco production has persisted, underpinning often successful opposition to tobacco control measures in the country [9]. This narrative has been mobilized domestically and globally [10] with only limited, and often anecdotal, evidence. Detailed analyses of the economic impacts of tobacco growing on farmer households presents a more nuanced picture that challenges this dominant narrative. First, tobacco growing is highly labour intensive. Ochola and Kosura [11] conducted a review of tobacco production in comparison to other crops in the Nyanza Province of Kenya and found that tobacco growing required an average of 220 labour days, dramatically greater than other crops such as passion fruit (less than 50), watermelon (50), soya bean (just over 50) and pepper (75). The labour requirements of tobacco growing require Kenyan farmers to enlist the support of family members (including children) and the wider community [12]. When these labour costs are included in assessments of household income, claims about the lucrativeness of tobacco growing become less tenable [3,11,12,13]. As with other crops, tobacco-growing households also experience massive variation in incomes from year to year. Heald [7], examining tobacco production in Kenya in the 1980s, observed a dramatic shift in incomes between 1984 and 1985. More recent research has shown that profits from tobacco production in Kenya are minimal given the input, labour, and other production costs [3,13]. Ochalo and Kosura [11] found that total costs, including input, opportunity, and hired labour, amounted to more than 65% of the average gross income of tobacco farmers. Magati and colleagues [14] surveyed tobacco farmers in Kenya in 2015 in order to analyze total profits from tobacco production, accounting for inputs and monetized value of family labour and found that “profits drop precipitously once labour is included: a US$13/acre net loss for contract farmers and a decrease in profit for independent farmers to US$43/acre.” These factors are particularly important given that Kenya is a country with profitable alternatives to tobacco in the tobacco-growing regions [7,8]. Importantly, recent research in Kenya finds that tobacco farmers are increasingly diversifying their income into other livelihood sources, finding that only a minority of tobacco-farming households rely on tobacco farming as their major income-earning activity [15].

There are many reasons why it is important to support a shift from tobacco farming to alternative livelihoods. First, as we indicated above, the labour requirements of tobacco growing are significantly higher than every other cash crop [3,14]. Second, tobacco farming poses several environmental stresses (e.g., water consumption, pesticide use, and deforestation from wood-fired curing), and health risks associated with ‘green tobacco sickness’ (acute nicotine poisoning) in leaf harvesting and smoke inhalation during leaf-curing [16]. Third, the Framework Convention on Tobacco Control (FCTC), the first public health treaty under the auspices of the World Health Organization, and to which Kenya is a party, calls on governments to promote alternatives to tobacco farming. 

However, there are several challenges to supporting such a shift. Despite price fluctuations, extremely high input costs, and deleterious health and environmental impacts, the tobacco supply chain, particularly for contract farmers, is much better structured than supply chains for other crops. Studies in Indonesia, the Philippines, Kenya, Malawi and Zambia indicate that farmers routinely express that tobacco is one of the only viable crops, stemming in considerable part from the receipt of extension services, inputs, and loans from leaf-buying companies, and the shorter distance to and existence of a predictable market with a guarantee of cash payment at the end of the season [6,17,18]. Thus, it is important to understand better the contextual factors that shape tobacco farmers’ decisions in order to target policies and programs that aim to shift farmers from tobacco to alternatives. 

To date, the literature exploring alternatives to tobacco farming has focused on national policy contexts and the views of farmers themselves [6,17,19,20], both important levels of inquiry. There is a need, however, to add the perspective of sub-national government officials and agricultural extension officers whose mandates in many countries are to support and guide agricultural production. This sub-national level is particularly important in Kenya, where recent devolution has shifted implementation of agricultural policies from the national government to subnational authorities (counties). In principle, this shift places greatly power in the hands of local agriculture officers to shape the landscape of agricultural production, which provides the administrative backdrop to understand the potential for alternatives to tobacco growing. Although there is some literature describing the challenges and benefits of tobacco growing in Kenya [3,6,14], there is limited data on how county-level government officials and agricultural extension officers at the sub-national level view tobacco growing, and the challenges to and opportunities for promoting alternative livelihoods. 

## 2. Materials and Methods 

This study used a qualitative description approach, a methodology best suited for “problem identification, hypothesis generation, theory formation and concept development” [21]. Qualitative methods emphasize the perspectives of those interviewed, allowing researchers to identify and describe themes and patterns across individual perspectives. 

Data was collected from two sources: semi-structured key informant interviews (*n* = 9) in 2019 and focus group discussions (FGDs) (*n* = 5 groups) in 2018. The two sources of data allow us to triangulate the perspectives of local government officials and other support workers with those of farmers who grow tobacco. Key informant interviews were conducted over a two-week period in May 2019 in the key tobacco-producing sub-counties of Suna West and Kuria West in Migori County and Teso North in Busia County. Informants included government officials at the county (*n* = 1) and sub-county (*n* = 5) levels. These informants occupied positions such as sub-county agricultural officials, agribusiness officers, and crops officers, all of whom work within the portfolio of agricultural policy and programs. Interviews were also conducted with extension officers (*n* = 3), whose activities include assisting farmers with issues and relaying information from the field to county government officials. Interviews were conducted in English by the lead author and two in-country research assistants, and followed one of two interview guides, with one guide used for extension officers and one used for all county and sub-county level government officials. The interview guides pursued questions regarding perspectives on tobacco farming, alternative crop farming, and the surrounding conditions affecting agricultural production, including the policy environment. The full interview guide can be found in Appendix A. Interviews with extension officers were approximately 10–15 min in length, while interviews with other government officials ranged in length from approximately 20 min to more than an hour. Interviews were conducted in government offices, restaurants, or homes, near or where key informants were located. Interviews were audio recorded and transcribed verbatim by the lead author. 

Focus group discussions were conducted between January and March 2018 in the three main tobacco-producing counties: Migori, Busia, and Meru. In Migori and Busia, two discrete groups were convened: current tobacco farmers and former tobacco farmers. In Meru, only one focus group with current tobacco farmers was conducted. Selection of participants for the focus groups was done during data collection of a parallel tobacco farmer survey in each of the regions. Enumerators, the individuals administering the survey to tobacco farmers, which is another component of our research project [6,14,18], identified 10–15 of the most experienced tobacco farmers in each county who were willing to provide additional information during the FGD. Efforts were made to generate some variation by gender, age, and the extent to which farmers had diversified livelihoods. FGDs were conducted by the principal investigator (PM) in both English and Kiswahili (local language) and transcribed verbatim into English. 

Interview and focus group transcripts were entered into NVivo qualitative software. Data were initially coded according to pre-identified categories derived from the research questions, and then reviewed to identify emergent themes and inductively coded into sub-categories. The lead author reviewed all data and conducted all coding under the supervision of RLen, and the final categories were mutually agreed upon. Ethics approval for this study was obtained from McGill University Institutional Review Board (IRB) (A09-E72-13A) and the Morehouse School of Medicine (the IRB of record for the American Cancer Society). 

## 3. Results

The viability of alternative crops and livelihoods depends on economic, institutional, and natural environments. Our findings illustrate the intersecting factors that shape the reasons for growing tobacco and the ability to pursue alternatives. First, we present findings on expressed benefits and stated challenges of tobacco farming. Next, we identify the stated challenges, opportunities, and suggestions for the future of alternative crop farming. Finally, we present participant reflections on the institutional context that shapes farmer practices. 

### 3.1. Tobacco 

The decision to farm tobacco was attributed to a variety of factors. We identified three categories of expressed benefits of tobacco growing: known market, profitability, and sustainable land use. The stated challenges are broken down into four categories: monetary losses, labour intensity, health risks, and environmental degradation. Additional representative quotes are presented in Appendix A.

#### 3.1.1. Stated Benefits

As with any crop, the decision to farm tobacco is variable and dependent on conditions in a given year. For example, the associated costs of farming in a given season can impact the decisions of farmers, as stated by a current tobacco farmer in Busia who explained that they would switch to tomatoes in the upcoming season “*since the cost of tobacco farming is high*”. A current farmer in Migori stated that:


*Everything has advantage and disadvantage, there is when the tobacco will help you and there is time you will experience loss. There is when you will have no water, or it gets burnt. At times you may experience loss and the insurance guys don’t pay you. At times you will get a good harvest and it will help you get good money. In the past there were good harvests but this is not happening nowadays.*


The decision to grow tobacco was repeatedly attributed to the ability of farmers to sell their crop and the money received in favorable seasons. Several participants in focus groups and stakeholder interviews expressed that this is possible due to the guaranteed market and lump sum payment methods that come with tobacco farming. An extension officer in Busia County said that *“If you talk to tobacco (farmers) what they believe is that tobacco is the market”*. Former tobacco farmers in Migori stated that *“Tobacco gave some good cash”* and that *“Tobacco had high undefeatable prices”*. A Busia County government official emphasized that, for tobacco, *“We see a future in terms of income generation to the farmers”*. The lump sum payment method utilized by leaf buying companies is attractive to farmers as it offers immediate cash payment. A current tobacco farmer in Busia County noted that *“By the time a person is leaving tobacco farming they have already (been able to make other) invest(ments) through tobacco”*. Examples of investments include starting a business, buying cattle, and building homes, all of which were seen to assist with future income generation. 

#### 3.1.2. Stated Challenges

Despite some study participants noting economic benefits from tobacco farming, there was a general consensus among governmental officers and current and former tobacco farmers that these economic benefits must be contextualized and interrogated, with participants consistently suggesting that tobacco is not a viable crop for farmers in their respective counties or sub-counties. A government official in Busia County stated that *“it is not a good venture, that is my stand”*, and many other officers echoed this sentiment. When asked if they would encourage their children to grow tobacco, a current farmer in Busia stated that he would *“not allow it”.* Another in the same focus group added that *“they have experienced the difficulties firsthand through their fathers”*. These difficulties were attributed to the small or non-existent profits received, the intensity of labour required, the health risks associated with production, and the environmental degradation caused by tobacco farming practices or the crop itself. 

We identified three reasons provided to explain the limited profits that farmers received from tobacco growing. The first was that there is often either no, limited, and/or delayed payment. Farmers expressed that they often are only able to clear the loans that they have taken with the companies, and after that must, as one current Busia farmer noted, *“keep the remaining tobacco and wait for the next selling season”*. Some participants indicated delay in payment by some leaf-buying companies of more than one year. Participants also claimed that the system used for grading can be easily manipulated to encourage profit for the company at the expense of the farmers. An extension officer in Busia County noted that farmers “*There are almost 10 grades of tobacco (and farmers often) don’t get the grade (they’re) expecting*.” To improve the grade of their tobacco, farmers often have to pay a bribe to the graders, as a current farmer in Busia stated, “*you have to bribe once you get inside there, this is Kenya, you give out money to get money*”. The third reason behind limited profits was that costs of inputs frequently exceeded the payments that farmers received. Farmers often must pay for firewood, additional labour, and other inputs out of pocket. A current farmer in Busia stated that *“You can be paid 200,000 shillings (~1900 USD) but when you calculate the expenses they add up to half a million (~4900 USD)”*. 

Additionally, in a related dynamic, the inputs farmers receive on loan from the leaf-buying companies can carry inflated costs, as told by a current farmer in Meru: “*When you go to the agro-input dealers you find the company inputs are highly priced than the other inputs*”. A current farmer in Meru County summed up the challenge famers must overcome when deciding whether to grow tobacco or to pursue alternative crops: 


*When you grow things like maize or beans you will never get cash, but when you plant tobacco with BAT you will sell it collectively, after working for a long time at the end you will get cash in lump sum. It may seem to be a lot of money but when you deduct the expenses, there is little left.*


The intensity of labour required to effectively farm tobacco can carry unfortunate side effects. It can limit other activities that the household could otherwise pursue (creating an opportunity cost). A government official in Busia County noted that farmers often *“spend a lot of time in the tobacco field at the expense of cultivating (their) food crop”*. This can leave farmers spending the little money they earned from tobacco on basic food items. Farmers also employ their children as a cost cutting measure, but as a former Busia County farmer notes *“absenteeism sets in … if you depend on the kids the performance will be down”*. 

Although our methodology cannot verify objectively that the health challenges experienced by tobacco farmers were attributable to tobacco farming, the narratives expressed by participants correspond with existing literature on the health risk of tobacco production. The curing process utilizes smoke and heat, which requires wood fires. Breathing in smoke from these fires can cause problems such as cancer or asthma and other chest problems. This can add to the financial burden on tobacco-growing families, as they must spend *“a lot treating diseases, among the family members”* (former farmer, Migori). The wood required for the curing process is the major cause of deforestation in tobacco-growing regions [16]: *“Before the onset of tobacco farming our country had large tree cover, currently the land is bare”* (current farmer, Migori). When trees on their own farms are depleted, farmers will often pay for wood fuel from neighbouring farmers, leading to further deforestation in the region. This finding is important given that all tobacco grown in Kenya is flue-cured. 

### 3.2. Alternative Crops

There is a wide variety of alternative (non-tobacco) crops that farmers elect to grow in the counties surveyed, however, several were mentioned repeatedly: cereals (sorghum, millet), tubers (sweet potatoes, cassava), vegetables (tomatoes, onions, beans, etc.), and horticultural products. All carry their own unique requirements, but the reality of pursuing these crops as an alternative to tobacco can be summarized into challenges and opportunities. 

#### 3.2.1. Challenges

The primary problem for farmers with non-tobacco crops is the lack of a reliable market and/or the often oversaturation of these markets, leading to uncertainty as to whether they can sell their crops after harvest. Farmers noted they must sell to brokers or risk finding their own customers. This limits their ability to plan financially and can limit their bargaining power when asking fair prices for their crops: *“If we had a known specific buyer we wouldn’t have the issue of fluctuating prices. Since we don’t have a specific buyer, people come and take advantage”* (former farmer, Migori). Markets have been established for some crops. One example identified by participants is the National Cereals and Produce Board, which attempts to organize the supply chain for maize. However, according to many of the farmers in our FGDs, the formal system does not function as it is meant to and farmers must resort to offloading their crop for low prices. As one current farmer in Migori explained: *“when you go to the National Cereal and Produce Board you will find maize spread outside because there is no space, you go back to the middleman who buys at any price”*. 

These low prices are exacerbated by the import of staple agricultural products: “we have situations where the government purchases food from outside, yet our food, we have it in plenty… whenever maize comes from outside, it floods the market and our farmers are demoralized” (government official, Migori). Two of the three counties surveyed, Busia and Migori, share borders with Uganda and Tanzania, respectively. Participants noted that because of the proximity to these two countries there existed a constant illegal movement of goods along with the legal importing of cheaper crops. Although these neighboring countries also provide another market for crops, in general it was expressed that there is a net influx of agricultural products into Kenya, most of which “is not going through the right channels, it will likely bring us problems” (Government Official, Busia). This influx challenges the ability of farmers to sell staple food crops at a price that is viable. 

The other commonly mentioned problems plaguing farmers are pests and diseases. Fall armyworm is considered a menace to farmers and has been destroying maize harvests for years in the counties surveyed. Government officials are attempting to *“sensitize farmers on the upcoming diseases and pests and how they can manage those challenges”* (government official, Migori) by providing trainings and assistance from extension officers, as well as the chemicals necessary for eradicating pests. Research organizations within Kenya work with the officers to promote disease resistant crops. For example, there is now a variety of cassava that is resistant to two serious diseases, cassava mosaic virus and cassava brown streak. Biotechnology, while thought to be useful for disease and pest eradication, *“has issues”*, a major one identified being Kenya’s tight restrictions on genetically modified (GM) crops. As one informant noted, *“In Kenya we haven’t come up with a policy of accepting that (biotechnology)”* (government official, Busia). 

#### 3.2.2. Opportunities

There are opportunities for alternative crops to yield higher or equivalent returns to tobacco, in part due to low barriers of production (please see representative quotes in Appendix A). Crops like maize and sweet potatoes are far easier to grow than tobacco. They take shorter times in the field, allowing farmers to harvest twice in a year. Compared to tobacco, the costs of production are low with most other crops, as they require fewer inputs and are less labour intensive. One farmer noted that *“tobacco expenses are thrice that of maize-growing expenses”* (former farmer, Migori). For maize and sweet potato, farmers must only weed once or twice during the growing season. Although according to some farmers these crops more typically yield lower prices, they can return enough profit to pay for farmers’ necessities, such as school fees. Another former farmer in Migori even mentioned that, *“For us who engaged in other crops, we have developed a lot. Some have even bought motorcycles for bodaboda (motor cycle taxi) business and we won’t even think of going back to tobacco farming again”*. 

Most participants (both government officials and farmers) noted the importance of pursuing value addition as a way to create local markets for certain crops: *“If we had a factory here in Kuria it would help the farmers in this region”* (former farmer, Migori). In order to maximize benefit to the farmers, a government official in Busia noted that it is important to advocate for alternative crops that *“will be processed here and most of the money will remain with the growers”*. There are several ways officers have been pursuing value addition in their regions: cottage industries, contract farming, and large-scale processors. A Migori County government official spoke of the importance of promoting cottage industries to farmers as these industries *“generate income in our society at their level”* and create markets. Implementing the tobacco model of a catered value chain for other crops could also prove beneficial. In Busia *“EABL, East Africa Breweries Limited, is contracting farmers to grow barley”* (government official, Busia), although this reference to crops for alcohol rather than food production raises an ethical issue of whether alternatives to tobacco production should similarly be considered for their possible negative impacts on health and social well-being (or even the environment). Larger companies are also providing a consistent market for products. One agricultural officer in Kuria noted that supply chains, including processing and product development, are being developed within Kenya, noting that:


*We also have processors; those are doing value addition. … they take this sweet potato, they puree, the pureed product they transform to biscuits, cakes, chapattis, and also bread. So these are products which are made from sweet potato. These products, the chips, the crepes, so many. And also the leaves can be eaten as a vegetable, very good in iron, very rich in vitamins. So the whole crop is very important. The peels can also be given to the livestock as treats.*


### 3.3. Institutional Context

The institutional context that farmers must navigate influences their economic livelihood decisions. Tobacco is an unscheduled crop in Kenya meaning that the government does not monitor (e.g., production records) or support (e.g., through input subsidies) tobacco production. Extension officers only see tobacco farms when they are attending to other crops on the same property. Officials and officers recognize the problems that farmers face due to tobacco (as indicated in the above sections), but there is limited government engagement with farmers to address these challenges. County-level government officials rarely receive information from the leaf-buying companies: *“We don’t know their role, we don’t know the volumes that they get, we don’t know the health implications, what they’re doing to mitigate them”* (Government Official, Busia). Apart from county governments having no ‘official’ role in tobacco monitoring or production, the lack of information from leaf-buying companies creates an additional barrier to tobacco being considered at all in county-level goals for agricultural production, specifically the goal of moving away from tobacco to other crops. It also gives them little recourse for discussing the hazards of tobacco growing with farmers. 

In order to encourage the production of alternative crops, officers mentioned policies and programs which provide free and subsidized inputs for low-income farmers. Officers in both counties surveyed mentioned the Farm Input Access Program (FIAP), which provides fertilizers and seeds to farmers for free: *“These are policies we have developed so that we can mitigate poor production, to cushion farmers against increasing inflation”* (government official, Migori). In practice, however, the farmers who most need access to these subsidized or free inputs might not receive them. An agricultural officer in Busia stated that the FIAP funds are being managed by Members of County Assemblies (MCAs) rather than the director, and *“with the Members of the County Assembly (MCAs) the main beneficiaries are their family members and close allies”*. An extension officer in Busia noted: 


*It was we the agricultural officers who were supposed to recruit the farmers and give them maize and fertilizer. But it’s now political… It is supposed to get to the poor. But you find if you are a chief, if you are a ward administrator, definitely your family will get it. It is not ending in the right place.*


Extension work is the most concrete way for information from government offices to reach farmers; this linkage is essential for the promotion of alternative crops and livelihoods. Extension services are supposed to be available upon farmers’ demand and are offered *“through field visits, farm visits, and also group approach”* (government official, Migori). These services are only possible with adequate funding, as officers must travel to individual farms to work with farmers. After devolution, extension services have been reduced or weakened. A Busia County government official stated: 


*We don’t have the platform to go and share it out because we are not facilitated, extension is not facilitated. The county is killing extension.*


Funding has become more difficult to obtain: “even the sub-county agricultural officer has no AIE (Authority to Incur Expenditure) … so now this has really frustrated extension” (government official, Migori). This means that many officers must pay out of pocket for vital resources such as fuel, motorbike repairs, and even pens/paper, limiting the work extension officers can undertake. In some cases extension officers may require payment from farmers themselves, as a Migori County current farmer stated: “if you want the extension officer from the Ministry of Agriculture to visit your farm you have to cater for the bills”. This can complicate efforts to reduce tobacco growing. Guidance is guaranteed when farmers grow tobacco as the extension officers are hired by tobacco companies. If farmers choose to grow an alternative crop, they risk losing assistance from experts as two tobacco farmers (one former, one current) noted: 


*In the past tobacco farmers were well established but after leaving tobacco, we have never seen any agriculture officer come to the farm to instruct us how to farm, we struggle on our own. (former farmer, Migori)*



*At least the tobacco farming has the technicians, the other farming of things like maize, beans you work it out on your own. (current farmer, Meru)*


## 4. Discussion

There was consensus among farmers and government officials across the counties represented in this study that the challenges of pursuing alternatives to tobacco production are extensive and the prospects of switching to alternative crops multifaceted and difficult. Although the monetary and physical costs of tobacco production are high, such as the harms to health and environment [16], the structured supply chain of tobacco incentivizes production. What we see are decisions that are primarily based on the structured supply chain and corresponding benefits of contractual relationships with leaf-buying companies, such as provision of inputs and access to loans. This is becoming a ubiquitous theme in studies that have examined why farmers choose to grow tobacco irrespective of the geographic location [6,17,22,23]. 

There exists a deep entrenchment and internalization of the dominant narrative regarding the economic benefits of tobacco at the macro level. For example, we often hear national governments in tobacco-growing countries defending tobacco production and opposing tobacco control measures based on this narrative [10,24,25]. Latest analysis (2016) presented in the district agricultural reports confirms that profits per acre for tobacco are much lower than other agricultural commodities in each of the three districts. In Busia, tobacco ranks 11th (24,585 shillings/acre), 9th in Meru (45,675 shillings/acre) and 6th (36,200 shillings/acre) (1 USD = ~100 Kenyan Shilling). To illustrate the difference between tobacco and other crops, a crop like tomato yields profits between 84,360–134,900/acre, groundnuts fetch a profit 54,360/acre, and maize between 35,450–76,760/acre. Interestingly the comments from county-level officials express a greater awareness that tobacco farming is not a profitable venture. Whether this awareness is due to proximity to the actual lives of farmers, or whether this is an indication that the tobacco industry tends to focus its efforts at the national level, is a matter of further investigation. Even among the current tobacco farmers who participated in this study, and who still repeat the story that tobacco is beneficial, there is widespread acceptance of its limited viability and its many harmful consequences. Farmers accept that it is an imperfect crop but will often justify their decisions to continue to grow tobacco based on the lack of alternatives. 

Farmers and government officials alike described the situation surrounding alternative crops as lacking, citing a lack of guaranteed markets, low prices, lack of oversight and direction on the part of the government as reasons for the reduced viability of alternative crops. Participants noted that an increase in agribusiness ventures in the counties surveyed could encourage farmers to switch from tobacco to alternative crops. Importantly, some farmers noted that they view tobacco growing as a short-term instrumental endeavour that will lead to other economic opportunities, such as business ownership. However, the actual ability of farmers to save money from tobacco farming to invest in other economic activities is limited. It is common for tobacco farmers to be caught in a cycle of debt, which forces them to continue to grow tobacco. This is perpetuated by a common practice by tobacco companies of refusing to buy tobacco from farmers once their loans have been reclaimed, the consistent downgrading of tobacco, and the inflated cost of inputs. Smallholder farmers of any crop can face this cycle of debt if the cost of inputs, typically in loans that might increase yearly, outweighs annual income [26]. Many finance institutions view smallholder farmers as high risk often leaving leaf-buying companies as the sole source of financing for farmers to access capital [27]. Again, the issue of household access to capital through loans or other means becomes a critical dimension when discussing alternatives to tobacco. This issue is part of a wider discussion of how to facilitate opportunities for alternatives to tobacco while at the same time ensuring fair prices for tobacco leaf to allow farmers to repay loans and have money left over for a sustainable economic livelihood [28].

Yet farmers and government officials noted that, in many cases, alternative crops can be more profitable and less hazardous than tobacco. Government officials have observational, anecdotal and experiential knowledge of the problems that many studies quantify, but there are many barriers to the pursuit of alternatives. Government participants noted that although they attempt to guide farmers towards alternative crops, they do not have the resources or even the official capacity to engage in this way. Unlike the extension services provided by tobacco companies, government extension officers are less able to form meaningful relationships with farmers, and thus have less impact on the decision-making processes of their clients. Moreover, data used to promote alternative crops are often based on hearsay and can even be influenced by (misleading) data from tobacco companies themselves, constraining country officials’ and officers’ abilities to promote alternatives based on economic or development arguments [9], [29]. 

There has also been a general shift within Kenya, and elsewhere, towards greater private sector management of agricultural supply chains for ideological (i.e., market-driven governance), monetary (e.g., reducing government budgets), and instrumental (e.g., incentivizing good agricultural practices) purposes [30,31]. Regardless of crop, government management of agriculture is receding as private sector engagement rises. In many cases the emphasis on reducing government involvement in agricultural supply is encouraged or even required by international economic agencies such as the International Monetary Fund [32]. The challenge of this withdrawal of government support to smallholder farmers is compounded by the influence of tobacco companies over communities. Tobacco companies dominate the economic landscape of smallholder tobacco farmers [9,33]. Although leaf-buying firms are not transparent about the size of their operations, recent survey research suggests that BAT is the dominant contracting firm, whereas the local firm, Mastermind, has a sizeable presence in one or two areas [15]. Tobacco companies are deeply invested in corporate social responsibility efforts to buttress their reputations and BAT even directly cites its positive contributions to meeting the UN’s Sustainable Development Goals [34]. Tobacco companies have a long but also recent history of seeking to undermine tobacco control policy, with a recent BAT whistleblower reporting to the United Kingdom government that he had bribed lawmakers across East Africa to oppose proposed legislation and regulation [35]. Farmers need information that alternative crops are economically viable and support to produce these crops, while the ability of government to provide this remains weak. Focus group participants often directly compared the benefits and challenges of tobacco growing to those of alternative crops. It was common for farmers, for example, to compare labour requirements and other inputs to that faced by community members growing other crops, often by listing what their neighbors were able to procure with their earnings. This signifies that farmers conduct their own personal cost-benefit analysis of crops, based on the minimal information available to them, and must be shown that alternative crops offer more concrete benefits.

In semi-arid regions of Kenya, production risk factors, including yield variance and downside risk, are important factors which influence adoption of new technologies. Farmers are traditionally risk adverse [36], and will generally maintain course unless circumstances are drastically worse in a given year. Technology adoption is unlikely to occur as poor farmers are unlikely to invest in anything that involves a possibility of downside risk, even if it promises a higher return than their current model of operation. Farmers who are less risk adverse generally see lower probabilities of crop loss occurring [36]. In order to encourage farmers to cease tobacco growing, programs, governmental or otherwise, must demonstrate that farmers can turn a larger profit and experience fewer hardships while growing alternative crops versus tobacco. 

Market factors weighed heavily in the discussions on the challenges of alternative crop farming, from both the FGD participants and government officials. The key factor identified was supply chain development, which hinges on market access and availability. Currently, most alternative crops are marketed very minimally; farmers must find buyers or markets themselves. Several officials spoke to ongoing initiatives targeting increased marketing of crops, although farmers did not mention similar programs. Government officials noted that the Ministry of Agriculture is placing a much larger emphasis on agribusiness, however, “Weak institutional frameworks discourage effective involvement in commercial agriculture” [37]. Agribusiness itself presents challenges to pursuing alternatives, as the development of new cash crops might repeat institutional risks associated with tobacco farming. Pursuing a single buyer for crops, for example brewing companies or supermarkets, can result in price taking by farmers (i.e., farmers have no control over the price given), minimizing profits and leaving them prone to price shocks [18].

An avenue to increase crop revenue mentioned by a government official is biotechnology, but this is a new concept in Kenya, lacks a legal and institutional framework for large-scale intervention [38,39] and, as such, is not yet accepted by Kenyan farmers. One biotechnology that is slowly being introduced into the Kenyan market is genetically modified (GM) crops, which are thought to improve crop yields. However, there has been little adoption of GM crops in the food supply, evident by the small number of approved field trials of GM maize and cotton. GM crops bring another layer of controversy and complexity in terms of corporate control, fair governance and farmer protection, which requires ongoing interrogation [40].

A major finding of this research is that the agricultural ministries in the counties surveyed are severely understaffed and underfunded. This makes it increasingly difficult for officers to execute useful activities to help shift farmers away from tobacco: establish markets, demonstrate profitability/other benefits of alternatives to tobacco, and procure and distribute fair-priced, affordable inputs. Government officials interviewed further noted the department operates via a top-down (national to county) approach, a claim supported by other research, an approach which is additionally described as uniform and inflexible [41,42]. This approach makes it difficult for county-level officials to engage with new problems as they arise or to allow staff to meet the requests of the farmers within their catchment areas. Government officials noted that decentralization diminished the efficacy of extension services as it limited their authority to incur expenditure. In cases where political decentralization does not come coupled with fiscal resources and authorities, the benefits typically associated with devolved decision making (increased accountability, increased care paid to needs of a local population) may not be realized [43], devolution processes interrupted key management structures of extension services, which limited accountability mechanisms and hindered the ability of staff to receive and integrate feedback from communities into their services. Programs meant to assist farmers and encourage a change in livelihood to alternative crops, along with other state provided services, may be prone to elite capture (e.g., via the MCAs, as noted by one of our interviewees) and often may not actually reach the intended beneficiaries. 

## 5. Conclusions

Tobacco farming is deleterious to human and environmental health and provides a tenuous economic livelihood for smallholder farms. Efforts to promote the replacement of tobacco to alternative crops at the sub-national level face a variety of challenges, which must be addressed in order to limit tobacco production. This study identifies challenges (monetary losses, labour intensity, health risks, environmental degradation) and benefits (known market, profitability, sustainable land use) of tobacco production as well as the challenges and benefits of alternative crops. The findings illustrate the intersecting factors that shape the reasons for growing tobacco and the ability of farmers to pursue alternatives by turning to an analysis of the institutional and policy context that impacts agricultural production. We find that issues of market access, availability of education and support for farmers and access to loans and other inputs are key factors in both continuing to pursue tobacco production or moving to viable alternatives. The research highlights institutional shortcomings, and the impact this has on agricultural production. This research can inform further analysis of the factors that shape agricultural production, the barriers and opportunities to shifts away from tobacco to non-tobacco crops and the wider policy landscape that, often inadvertently, perpetuates tobacco production.

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
