# Peer review of "Understanding Alternatives to Tobacco Production in Kenya: A Qualitative Analysis at the Sub-National Level"

_ijerph, 2020, doi:10.3390/ijerph17062033_

Round 1
Reviewer 1 Report
The authors discuss findings from data obtained through discussions with farmers and regional level stakeholders in Kenya's tobacco growing areas. Farmers' experiences and perceptions of alternative crops are presented along with analysis of the factors that influence alternatives to tobacco production. Overall, the analysis is an important contribution to discussion about tobacco control and agricultural production in Kenya and other non-dominant tobacco growing countries. I learned a lot and look forward to reading the published manuscript. I have a number of minor questions and concerns.
The richness of the manuscript is the set of responses from farmers and former farmers. It seems the voices of farmers are not necessarily used in an optimal manner. Up front, rather than waiting till page four, readers may want to learn about an actual farmer and his experiences through an anecdote or vignette. The presentation of interview findings in short sentences in the bulk of the article as opposed to longer excerpts seem to demonstrate insufficient use of qualitative information. I'm not convinced that relegating much of the quotes from interview findings in the appendix is helpful for the project. Might be useful to integrate on an needed basis the interview excerpts in the main text to provide readers with a direct connection to farmers' responses. Though, i recognize this may be a discipline-related issue.
We learn that 55,000 farmers grow tobacco. Are all of these farmers contract farmers? According to British American Tobacco Kenya's 'About Us' page, the company contracts with 5,000 farmers in Kenya. Additional details on the types of farmers and contract and non-contract arrangements.
Are farmers only growing flue cured tobacco which requires large amounts of wood for curing? Or are there other types such as burley being cultivated which requires significantly less wood for processing purposes?
The authors do a nice job discussing economic challenges and consequences of tobacco production. But i think they gloss over too much the economic benefits of tobacco farming. A more well-rounded discussion might provide a more nuanced approach to what the benefits of tobacco production are in the study areas, especially since the cultivation of tobacco will likely persist in Kenya for a generation or more.
Discussion of companies that dominate the tobacco supply chain is missing. If BAT only contracts with a small portion of farmers what leaf buying companies or entities purchase the bulk of tobacco leaf? I understand BAT operates a foundation in Kenya and administers corporate social responsibility projects in tobacco growing regions. The company boasts about traceability to the farm level in Kenya and even supports crop diversification for tobacco farmers (see p17 in BAT's 2018 Sustainability Report). BAT activities show that it has tentacles in some local areas and likely exerts influence along economic and political lines. This is an area that needs some discussion especially because the company's demonstrated track record to appropriate meaningful terms and use them to undermine genuine efforts to exit tobacco farming and seek alternative opportunities.
Shillings, the currency in Kenya, are mentioned. Readers may benefit for conversion to dollars or an equivalent.
While it may be true that 'wood required for the curing process is a major cause of deforestation in tobacco growing regions' (p5, line 234)- a citation is needed, in my humble view.
On page 9 line 419 some miscellaneous text needs to be deleted.
I am skeptical of genetically modified crops being mentioned in any discussion of alternatives only because there are some that believe this to be a contemporary form of recolonization via agricultural production. I believe that the area of GMOs is associated with some of the dominant funders in global tobacco control and it's concerning to me and maybe for other readers. Going deeper, maybe use the space to recognize the point in the December 2019 piece "International Labour Organization (ILO) ends tobacco industry funding" that tobacco control researchers and global public health educators "Support alternative livelihoods, while promoting fair living earnings for tobacco farmers and farm workers and fair tobacco leaf prices."
In the references in item 7, the authors cite 'factfish tobacco.' it may make sense to add a note about how this source contrasts (is an improvement upon?) the FAOSTAT database of unmanufactured tobacco.
Reviewer 2 Report
This article offers in many senses a welcome and nuanced approach to addressing the issue of tobacco production and its persistence in Kenya. It interrogates a clear gap in wider literature, focusing on in-country governance structures and how they represent barriers to the shift from tobacco to other crops, and draws on a wide range of research to present a broad view of tobacco cultivation and its issues.
I would recommend however that some key issues be addressed before this article is accepted for publishing, as follows:
A key data point that would be useful: what proportion of tobacco farmers’ household income comes from agriculture generally? In rural small farming households across this region, agricultural income is declining as a share of household income, so to what extent should the state be enabling tobacco farmers to transition out of agriculture altogether, as opposed to into new crops?
- There is a continual reference to farmers’ ‘perceptions’ or ‘perceived benefits’, contrasted by seemingly objective research from studies showing the opposite. This binary suggests that farmers’ own knowledge is somehow less robust, and the tone of the piece therefore becomes about researchers teaching farmers to move away from tobacco, which does little to understand why tobacco remains so popular. I think it is important to afford more validity to farmers’ knowledge, and so would suggest altering ‘perceptions’ to something more concrete. Without this, the study falls into a long-standing trap of treating farmers’ knowledge as subordinate, rather than seeking to understand and work with them.
- Lines 68-88: the literature review indicates that tobacco is less profitable than suggested through examining issues with it, such as labour requirements, price fluctuation, etc. Later (l.90) it is indicated that labour requirements making tobacco ‘economically precarious’ is a reason to switch from it. However there is no contextualisation with other similar crops – coffee, cocoa, cotton etc, many of which suffer the same issues as indicated for tobacco. Without this, the arguments against tobacco weaken, as they are arguably general arguments against the issues of cash crop production under a neoliberal export regime, and can apply to many crops. Perhaps focus on issues specific to tobacco? Green tobacco sickness, its ill-health effects… I would suggest that this section needs to either dial back the rhetoric about tobacco being exceptional in terms of issues raised, or in keeping the rhetoric, include problems more specific to tobacco.
- In understanding the economic benefits, do you have any quantitative data for tobacco and other crops, even just yield/profit per hectare estimates? Without any data, the arguments around tobacco costing more lack substance, as they are based on seemingly contradictory notions of it being both profitable and costly.
- The ability to cure and store tobacco is something that separates it from other crops, in giving farmers the option of selling in low periods of the market (and so gaining higher prices), and also in enabling them to capture higher rungs of the value chain. This is not discussed here, it seems from the discussion on health problems that farmers do cure their own tobacco, but can you refer to this more in the section on economic opportunities?
- In the final discussion again (l. 359 onwards), there is a juxtaposition between the ‘narrative’ that tobacco is profitable, and the ‘reality’ that it isn’t. Yet this comes after a nuanced section highlighting problems with other crops as well. I would encourage you to alter the language around this, as we need to embrace the fact that in many instances tobacco is profitable in comparison to other cash crops, and start from this as the basis to get farmers to move away from it. Denying that tobacco is ever profitable seems to be a false basis from which to move beyond it.
- As well as under-staffing and under-funding of agricultural agencies (l.448-9), the authors may want to address the limits placed upon the government’s ability to provide assistance to farmers in cultivating export crops that result from regional trade agreements, or loan conditionalities from the IMF/ World Bank. The introduction of subsidies, state assistance, and extension services are often prohibited by these trade agreements/ loans, and seen as market distortions, and so the government cannot enable these, and has its hands tied.
